# Porosity Characteristics and Effect on Tensile Shear Strength of High-Strength Galvanized Steel Sheets after the Gas Metal Arc Welding Process

**Seungmin Shin** and **Sehun Rhee** *

Department of Mechanical Convergence Engineering, Hanyang University, Seoul 133-791, Korea;
glipzide@naver.com
* Correspondence: srhee@hanyang.ac.kr; Tel.: +82-2-2220-0438

**Abstract:** In this study, lap joint experiments were conducted using galvanized high-strength steel, SGAFH 590 FB 2.3 mmt, which was applied to automotive chassis components in the gas metal arc welding (GMAW) process. Zinc residues were confirmed using a semi-quantitative energy dispersive X-ray spectroscopy (EDS) analysis of the porosity in the weld. In addition, a tensile shear test was performed to evaluate the weldability. Furthermore, the effect of porosity defects, such as blowholes and pits generated in the weld, on the tensile shear strength was experimentally verified by comparing the porosity at the weld section of the tensile test specimen with that measured through radiographic testing.

**Keywords:** GMAW; zinc; porosity ratio; tensile shear strength

---

## 1. Introduction

Currently, many car makers are increasingly applying high-strength steels to meet weight requirements to increase economic efficiency and productivity. In particular, in the gas metal arc welding (GMAW) process, galvanized steel sheets are often applied to automotive chassis components owing to their excellent corrosion resistance and formability [1–4]. However, when lap joint welding is applied to galvanized steel sheets, high-pressure zinc fumes form, as the zinc coating is vaporized in the lap joint. If the high-pressure zinc fumes are not completely discharged outside, they remain in the molten pool and cause weld defects, such as blowholes and pits, inside and outside the weld [5–7]. These weld defects significantly reduce the durability and productivity of the weld [8,9].

Lee et al. investigated the correlation between porosity defects and fracture strength in the $CO_2$ welding of galvanized steel sheets and characterized the defects generated under different welding conditions [10]. Izutani et al. proposed guidelines to reduce porosity defects based on observations of the porosity formation behavior in the lap joint welding of galvanized steel sheets [11]. Yu and Kim analyzed the optimal conditions of the torch angle and aiming position for a lap joint weld during the cold metal transfer (CMT) process of galvanized steel sheets applied to chassis components, where they experimentally reduced the porosity in the weld [12].

As shown in these examples, many studies have analyzed the porosity formation mechanism and attempted to reduce the porosity in the weld [13–16]. Recently, Kim et al. experimentally investigated the effect of porosity on the fatigue behavior in the GMAW process [17]. To effectively discharge zinc vapor in the GMAW process using zinc-coated steel sheets, Kam et al. investigated controlling the porosity using gap-paste-based Ti particles between galvanized steel sheets [18]. However, few studies have quantitatively analyzed the correlation between the porosity and the tensile shear strength (TSS) of an actual weld and their effects on the GMAW process. In addition, no studies thus far have analyzed the porous components formed on a weld to verify the remaining zinc.

This study conducted lap joint experiments using the galvanized high-strength steel, SGAFH 590 FB 2.3 mmt, which is applied to chassis components in the GMAW process. Zinc residues were examined by analyzing the composition of the porosity in the weld and the weldability was evaluated through tensile shear tests. Furthermore, the porosity was measured by performing radiographic testing (RT) to ascertain the effects of porosity defects, such as blowholes and pits generated in the weld, on the tensile shear strength. Finally, the experimental results of the porosity at the welded section of a tensile test specimen were compared with that measured through RT.

## 2. Experimental Procedure

### 2.1. Materials and Experimental Method

In this study, SGAFH 590 FB 2.3 mmt, which is applied to automotive chassis components, was used. Table 1 lists the chemical composition and mechanical properties of the specimen.

**Table 1.** Chemical composition and mechanical properties of the specimen.

| Materials | Chemical Composition (wt.%) | | | | | | Mechanical Properties | | |
|---|---|---|---|---|---|---|---|---|---|
| SGAFH 590 FB | C | Si | Mn | P | S | Fe | YS (MPa) | TS (MPa) | EI (%) |
| | 0.0817 | 0.136 | 1.440 | 0.013 | 0.002 | Bal. | 583 | 629 | 25 |

The welding experiment was performed in the short-circuit transfer mode using a 450A-class constant voltage inverter DC-type welding machine (Fronius, Wels, Austria). Figure 1a shows the welding jig used in this experiment. The welding test specimen used in the tensile shear test after welding to assess the weldability were 180 mm wide and 150 mm long. Lap joint welding was applied with fixed overlaps of 15 mm at a torch angle of 45°, as shown in Figure 1b.

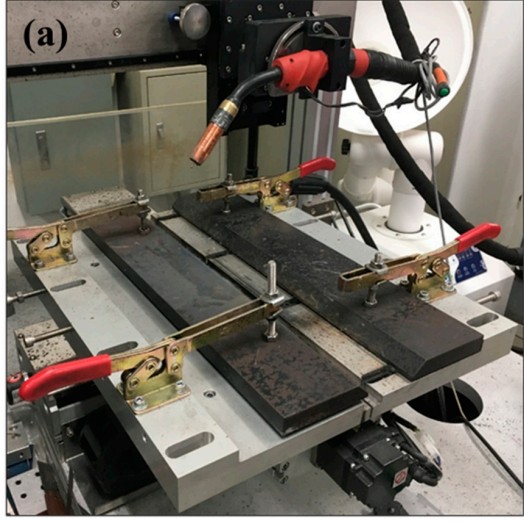
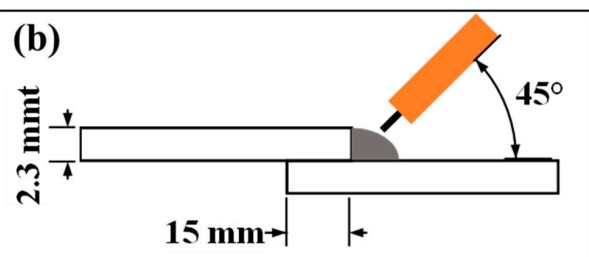

**Figure 1.** Welding experimental setup. (**a**) Welding jig; (**b**) welding type.

Figure 2a shows the universal tensile tester with a maximum load of 30 tons, which was used to evaluate the weldability. To measure the porosity ratio of the weld, RT was performed using an RT testing machine (XAVIS Co., Ltd., Seongnam-Si, Korea), as shown in Figure 2b.

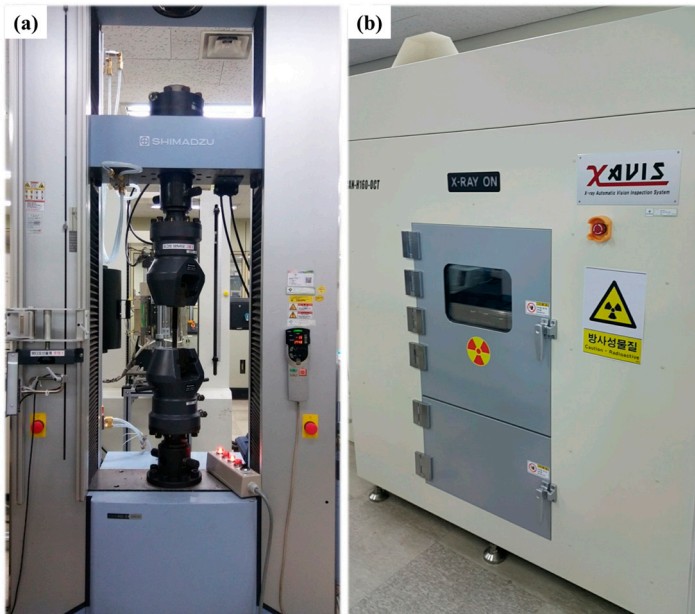

**Figure 2.** Test equipment. (**a**) Universal tensile testing machine; (**b**) radiographic inspection testing machine.

## 2.2. Welding Experiment Conditions

The welding experiments were performed with and without a gap in the specimen lap joint. The reason for experimenting with a gap in the specimen lap joint was to attain a good weld by providing space for discharging the zinc fumes generated from the zinc plating during arc welding, thus minimizing the weld porosity. The wire feed rate (WFR) and the contact-tip-to-workpiece distance (CTWD) were fixed at 3 m/min and 15 mm, respectively. The welding rate was 600 mm/min. For the protective gas, an Ar/$CO_2$ mixture gas at a ratio of 9:1 was used. For the welding wire, a $\varnothing$ 1.2 mm solid wire for high-tension steel was used. To ensure reliability, the experiments were repeated three times per condition. Table 2 lists the experimental conditions.

**Table 2.** Welding experimental conditions.

| Welding Conditions | |
| --- | --- |
| Gap (mm) | 0, 0.5 |
| CTWD (mm) | 15 |
| WFR (m/min) | 3 |
| Welding Speed (mm/min) | 600 |
| Shielding Gas | Ar (90%):$CO_2$ (10%) |

Table 3 lists the results of the experiments with and without the gap. In the absence of a gap, pits formed throughout the surface of the weld, whereas in the presence of a gap, no pits formed on the surface of the weld.

## 2.3. Weldability Assessment and Porosity Measurement

In this study, the tensile shear tests were performed to quantitatively assess the difference in the tensile shear strengths between a weld with pores and a weld without pores. As shown in Figure 3a, the specimens are processed in accordance with the specifications of the tensile test specimen no. 5 in KS B 0801 (test pieces for tensile test for metallic materials) [19]. Furthermore, RT was performed to measure the porosity ratio inside and outside the weld. As shown in Figure 3b, the tensile shear tests

were performed with three tensile test specimens per condition to obtain reliable data and the porosity ratio was measured for each tensile test specimen.

**Table 3.** Appearance of weld bead according to experimental conditions.

| Run | Current (A) | Voltage (V) | Weld Bead Appearance | Gap (mm) |
|-----|-------------|-------------|----------------------|----------|
| 1 | 113 | 18 | | 0 |
| 2 | 114 | 18 | | |
| 3 | 120 | 17 | | |
| 1 | 116 | 17 | | 0.5 |
| 2 | 120 | 17 | | |
| 3 | 117 | 18 | | |

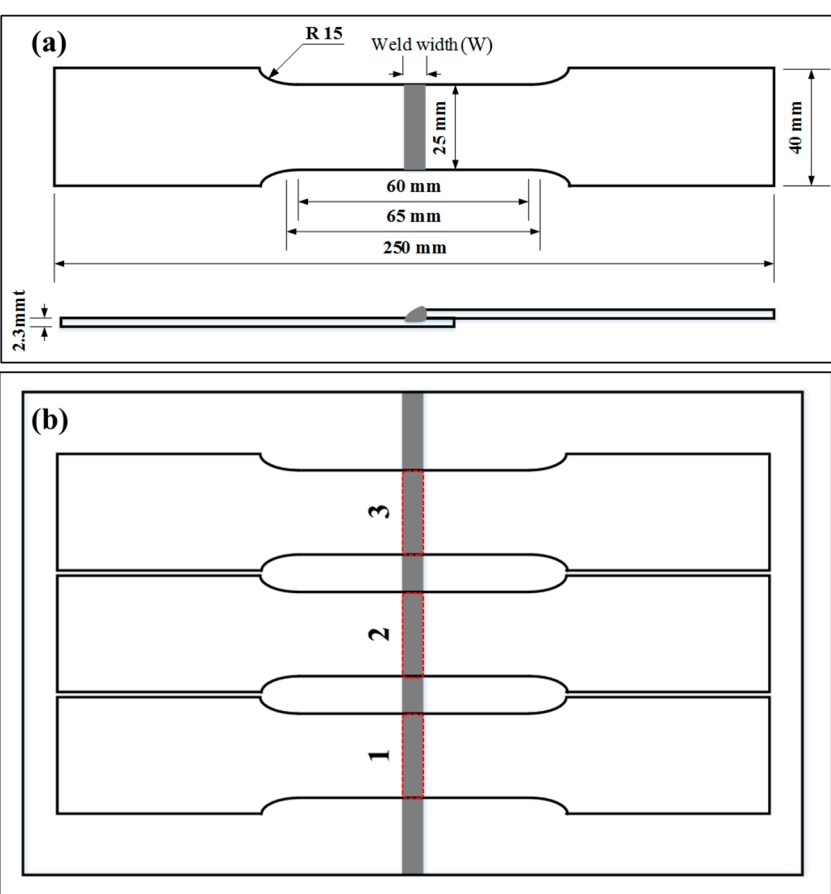

**Figure 3.** Specifications of the tensile test specimen for weldability evaluation: (**a**) tensile test specimen size; (**b**) tensile test specimen processing method.

To measure the porosity, assuming that several pores ($d_1$–$d_i$) are generated in the entire weld with a weld length $L$ and a weld width $W$, as shown in Figure 4, the total weld area is determined by multiplying $L$ and $W$, as shown in Equation (1).

$$Weld\ Area\ (mm^2)\ =\ L\ (mm)\ \times\ W\ (mm) \tag{1}$$

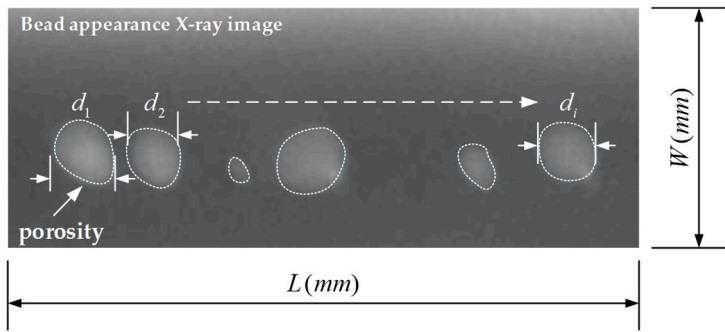

**Figure 4.** Total porosity area measurement.

Next, assuming that the pores ($d_1$–$d_i$) in the weld are circular, the total porosity area is determined by summing the areas of these pores ($d_1$–$d_i$), as shown in Equation (2).

$$Total\ Porosity\ Area\ (mm^2) = \sum Porosity = \sum_{i=1}^{n} \frac{\pi d_i^2}{4} \qquad (2)$$

Finally, the porosity ratio can be determined by dividing the total porosity area by the total weld area, as shown in Figure 3.

$$Porosity\ (\%) = \frac{Total\ Porosity\ Area\ (mm^2)}{Weld\ Area\ (mm^2)} \times 100 \qquad (3)$$

## 3. Results and Discussion

### 3.1. Effect of Zinc Component on Porosity

Zinc fumes form when the zinc coating is vaporized at the weld lap joint during the arc welding of a galvanized steel sheet using the lap joint method. If these zinc fumes are not completely discharged, porosity defects, such as blowholes and pits, are generated both inside and outside the weld. In this study, a semi-quantitative energy dispersive X-ray spectroscopy (EDS) (Thermo Fisher Scientific, Waltham, MA, USA) analysis was performed to verify the existence of zinc residues in the porous parts of an actual weld. Figure 5 shows the EDS analysis results, which verify the existence of zinc residues in a welded section where pores are generated. Figure 5a shows an 80× enlarged image of the welded cross section. Figure 5b shows a 300× enlarged image at point p0, where the region is porous, as shown in Figure 5a. Figure 5c shows a 5000× enlarged image at point p1, where the region is porous, as shown in Figure 5b. Consequently, 2.70, 4.30, and 5.02 wt% zinc was found to remain at points p2, p3, and p4, as shown in Figure 5d–f, respectively. Thus, the EDS analysis results confirm that zinc remained in the porous locations.

An elemental mapping and line scan analysis was performed to verify the existence of zinc residues at the porous locations in the weld. Figure 6a shows the image of the plated layer of the weld and the porous location. Figure 6b–f shows the elemental mapping analysis results. In Figure 6f, the purple image indicates the zinc component. Zinc remain even at porous locations that are not in the zinc coating.

In addition, Figure 7a–c shows the results of the EDS line scan analysis. Figure 7a shows the results from two pores scanned horizontally in the line scan. Figure 7b,c shows the vertical line scan analysis for pores 1 and 2 in Figure 7a. The results show that zinc remains in both pores.

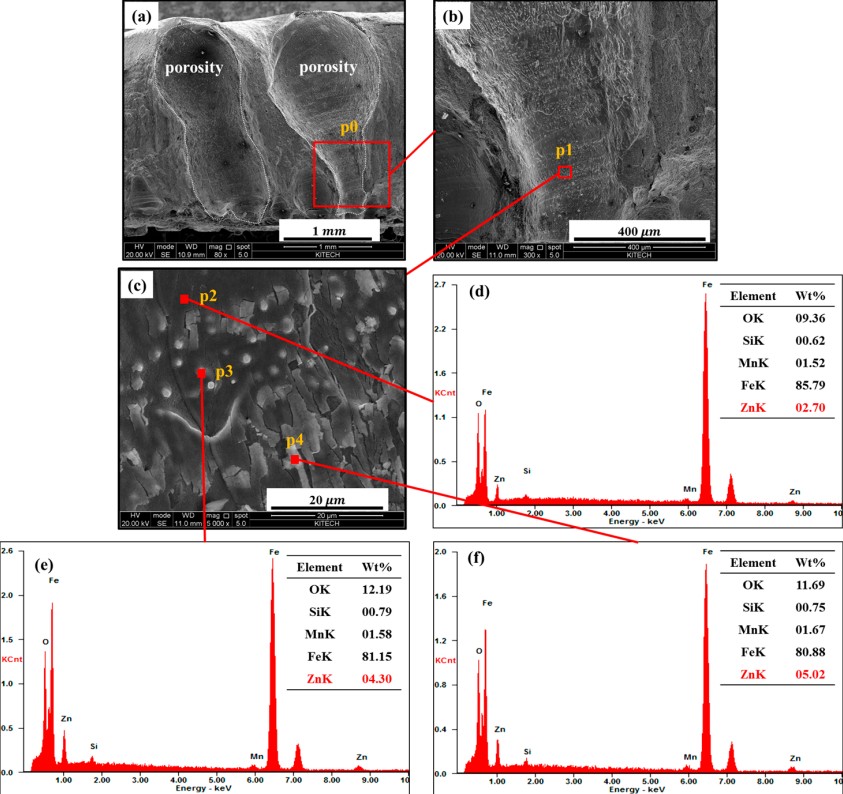

**Figure 5.** EDS analysis of porous locations in the welded section: (**a**) 80× image of the welded cross section; (**b**) 300× image at point p0; (**c**) 5000× image at point p1; EDS component analyses at (**d**) point p2, (**e**) point p3, and (**f**) point p4.

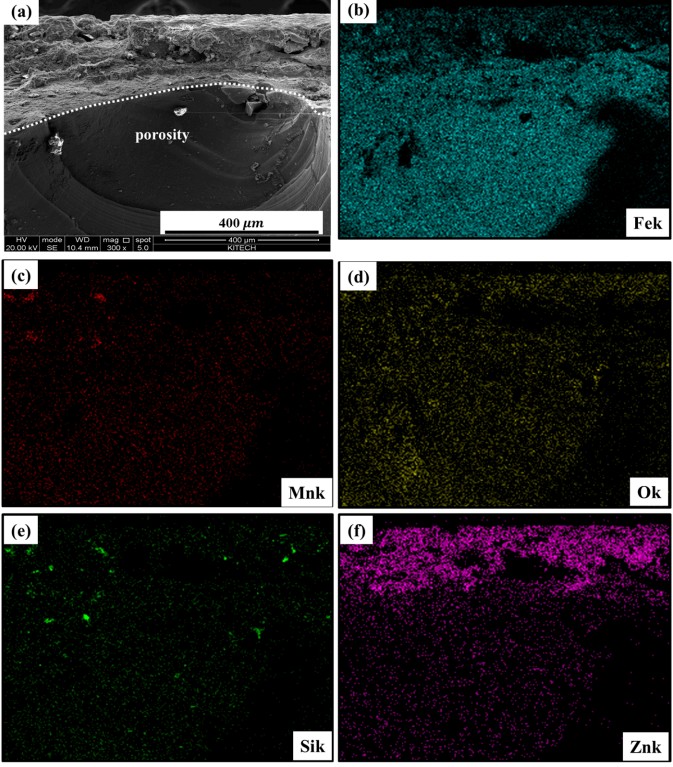

**Figure 6.** Elemental mapping analysis of porosity cross section of weld: (**a**) 300× image of porous location in the weld; (**b**) FeK, (**c**) MnK, (**d**) OK, (**e**) SiK, and (**f**) ZnK elemental mapping analysis results.

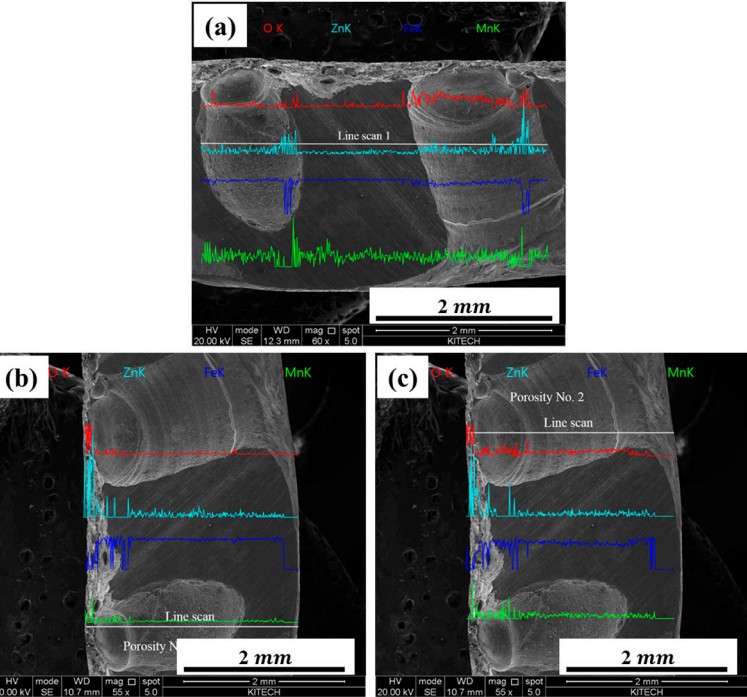

**Figure 7.** Energy dispersive X-ray spectroscopy (EDS) line scan analysis of porous locations in the welded section: (**a**) horizontal line scan image for pores 1 and 2 from the welded cross section; vertical line scan images for (**b**) pore 1 and (**c**) pore 2.

In addition, the existence of zinc residues was verified by performing a semi-quantitative EDS analysis of porous and non-porous locations in the weld. Figure 8a shows a 100× enlarged image of the welded section in which pores are observed. Figure 8b–d shows the semi-quantitative EDS analyses of the zinc coating as well as porous and non-porous locations in the weld, respectively.

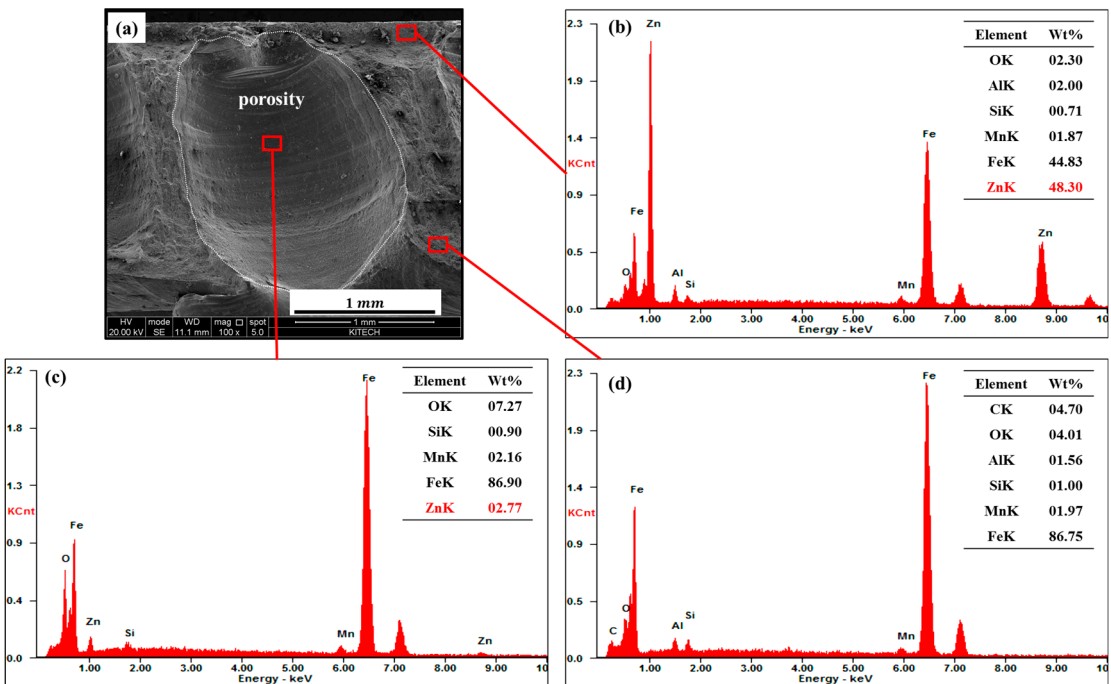

**Figure 8.** Analysis of EDS components at porosity welded and non-welded locations: (**a**) 100× image of the welded section; EDS component analyses of the (**b**) zinc coating, (**c**) porous location in the weld, and (**d**) locations where no pores are generated.

The analysis results confirm that 48.30 wt% zinc remained in the zinc coating, 2.77 wt% remained in the porous locations, and no zinc remained in the non-porous locations. These results confirm that zinc remained because of the zinc fumes at the porous locations.

### 3.2. Weld Porosity and Tensile Shear Strength

Table 4 lists the results of the experiments conducted with no gap in the lap joint corresponding to a welding rate of 600 mm/min and a WFR of 3 m/min. The weld bead surface of each tensile test specimen, X-ray images, and weld fracture shapes under different conditions are compared. The weld surface contains pits, and pores form inside the weld, even in the section where no pits are generated, as observed in the X-ray image obtained through RT. This is due to the high-pressure zinc fumes generated from the lap joint that were not discharged but remained inside the molten pool. Moreover, the tensile shear test results confirm the formation of weld fractures under every condition. This appears to be because of the decreased weld strength owing to the formation of pores inside and outside the weld.

**Table 4.** Weld bead surface and fracture shape of each test specimen (gap: 0 mm).

| Run | Item | Tensile Test Specimens | | |
| :---: | :---: | :---: | :---: | :---: |
| | | 1 | 2 | 3 |
| 1 | Weld surface | | | |
| | X-ray image | | | |
| | Fracture shape and mode | | | |
| | | Weld Fracture | Weld Fracture | Weld Fracture |
| 2 | Weld surface | | | |
| | X-ray image | | | |
| | Fracture shape and mode | | | |
| | | Weld Fracture | Weld Fracture | Weld Fracture |
| 3 | Weld surface | | | |
| | X-ray image | | | |
| | Fracture shape and mode | | | |
| | | Weld Fracture | Weld Fracture | Weld Fracture |

Table 5 lists the porosity and tensile shear strength values for each tensile test specimens, obtained using Equations (1)–(3). The formation of pores is considered one of the causes of the decreased weld strength. As listed in Table 5, in every tensile test specimen in which pores are observed, a higher weld porosity ratio led to a lower TSS with respect to the allowable tensile shear strength.

**Table 5.** Porosity and tensile shear strength (TSS) of each test specimens (gap: 0 mm).

| Run | Item | Tensile Test Specimen | | |
| --- | --- | --- | --- | --- |
| | | 1 | 2 | 3 |
| 1 | Porosity ratio (%) | 7.3 | 7.5 | 4.2 |
| | TSS (MPa) | 312 | 434 | 391 |
| 2 | Porosity ratio (%) | 2.7 | 8.0 | 5.3 |
| | TSS (MPa) | 463 | 267 | 337 |
| 3 | Porosity ratio (%) | 3.2 | 1.7 | 3.0 |
| | TSS (MPa) | 346 | 438 | 345 |

Table 6 lists the experimental results obtained for a 0.5 mm gap in the lap joint under the same conditions as those listed in Table 4. No pits are generated on the weld surfaces of the specimens. Furthermore, the X-ray images from RT show that no pores formed inside the weld. No pores are present because a gap was employed in the lap joint to allow the high-pressure zinc fumes generated from the lap joint to escape, thus minimizing the occurrence of pores in the weld. Moreover, the tensile shear test results show that the workpiece fractured under all conditions. This could be because good welds were attained, with no pores inside or outside the weld.

**Table 6.** Weld bead surface and fracture shape of each test specimen (gap: 0.5 mm).

| Run | Item | Tensile Test Specimens | | |
| --- | --- | --- | --- | --- |
| | | 1 | 2 | 3 |
| 1 | Weld surface | | | |
| | X-ray image | | | |
| | Fracture shape and mode | Base metal Fracture | Base metal Fracture | Base metal Fracture |
| 2 | Weld surface | | | |
| | X-ray image | | | |
| | Fracture shape and mode | Base metal Fracture | Base metal Fracture | Base metal Fracture |
| 3 | Weld surface | | | |
| | X-ray image | | | |
| | Fracture shape and mode | Base metal Fracture | Base metal Fracture | Base metal Fracture |

Table 7 lists the porosity and TSS values of each test specimen with no pores. The TSS of every test specimen is higher than the allowable tensile shear strength.

**Table 7.** Porosity and TSS of each test specimen (gap: 0.5 mm).

| Run | Item | Tensile Test Specimen | | |
|---|---|---|---|---|
| | | 1 | 2 | 3 |
| 1 | Porosity ratio (%) | 0 | 0 | 0 |
| | TSS (MPa) | 613 | 614 | 619 |
| 2 | Porosity ratio (%) | 0 | 0 | 0 |
| | TSS (MPa) | 620 | 621 | 622 |
| 3 | Porosity ratio (%) | 0 | 0 | 0 |
| | TSS (MPa) | 614 | 611 | 613 |

*3.3. Correlation between the Porosity Ratio and Tensile Shear Strength*

Figure 9 shows the correlation between the TSS and the porosity ratio in the weld with and without the gap based on the experimental results listed in Tables 5 and 7. These results confirm that the shear tensile shear strength decreases when pores are formed in the weld without a gap and when there are no pores with the gap, the shear tensile shear strength becomes higher than the allowable tensile shear strength value of 590 MPa. This suggests that weld porosity significantly affects the weld strength.

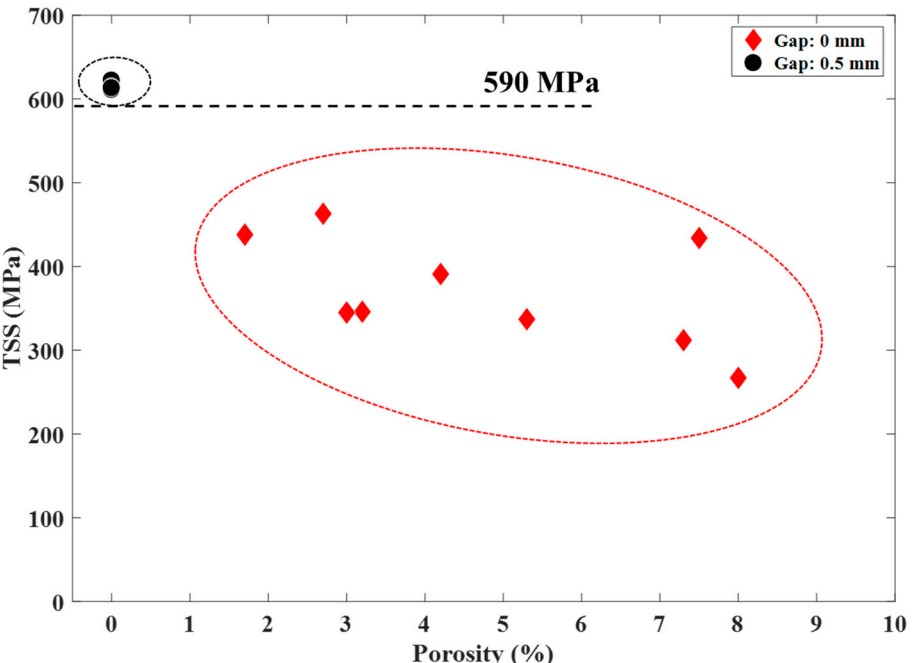

**Figure 9.** Correlation between the porosity and the TSS.

## 4. Conclusions

In this study, experiments were performed using SGAFH 590 FB 2.3 mmt, which was applied to automotive chassis components in the GMAW process. Moreover, an EDS analysis was performed to verify that weld porosity is due to zinc fumes that are not discharged outside (i.e., fumes that remain inside the weld). The weldability was assessed through tensile shear tests and the porosity was measured through RT to quantitatively assess the weld porosity. The following conclusions are drawn from this study.

(1) The semi-quantitative EDS analysis of the porous locations helped confirm the existence of zinc residues in general, but no zinc remained in the non-porous locations, which suggests that zinc fumes affect the formation of pores.

(2)　　A gap of 0.5 mm in the overlap of the lap joint method is an efficient way to discharge the zinc fumes, thus preventing the formation of pores in the weld.

(3)　　The weld strength was experimentally demonstrated to decrease significantly when pores were generated in the weld.

**Author Contributions:** S.S. performed the welding experiments, analyzed the data, and wrote the paper; S.R. reviewed the paper.

**Acknowledgments:** This work was supported by the Industrial Technology Innovation Program (No. 10063421, 'Development of the in-line welds quality estimation system and network-based quality control technology in arc and spot welds of ultrahigh-strength steels for automotive parts assembly') funded By the Ministry of Trade, Industry and Energy (MI, Korea). This research was respectfully supported by the Engineering Development Research Center (EDRC) funded by the Ministry of Trade, Industry and Energy (MOTIE) (No. N0000990).

**Conflicts of Interest:** The authors declare no conflict of interest.

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
