# Peer review of "Porosity Characteristics and Effect on Tensile Shear Strength of High-Strength Galvanized Steel Sheets after the Gas Metal Arc Welding Process"

_metals, doi:10.3390/met8121077_

Reviewer 1 Report

Review report

This work attempts to investigate the effects of weld porosity on tensile shear resistance of a galvanized steel joints obtained by GMAW. This paper can be suitable for publication in Metals only after a major revision. The results and discussion is quite superficial and also the conclusions. The authors did not explain which zinc compounds formed during the welding inside the porosities and which effects may have this compounds on shear resistance of the joints. Moreover, a clear correlation between TSS and porosity ratio increase cannot be observed because further tests should be made, in my opinion. The effect of porosity is quite obvious, but which influence of welding parameters was found on porosity formation and thus on TSS? The authors should improve these topics to give a wider explanation.

The English language must be improved in several parts of the manuscript. The following comments are given below:

Specific comments:

1.     Experimental procedure, paragraph 2.1: please correct the type of character of sentences from: “The welding experiment……….as shown in Fig. 1(b)”.

2.     Experimental procedure, paragraph 2.2: The welding rate unit of measure must be correct, I don’t think it could be mm/mindm.

3.     Experimental procedure, table 3: Maybe the term “Bead shape” is not very appropriate. Is the author meaning porosity distribution and shape in the welding? Please move the gap column after the voltage and adjust the width of the columns so that the words Current Voltage are on one line.

4.     Experimental procedure, paragraph 2.3: Please specify what is the parameter ɸdi (is the diameter of each pore?). The equations (1) and (3) are not legible.The porosity rate is the porosity ratio? How did you indicate it with R%?

5.     Results and discussion, paragraph 3.1: It is better to write “EDS semi-quantitative analysis” in line 4. What does the author mean with Zinc components? Elemental Zinc or Zinc compounds (maybe like intermetallics componds or zinc oxides)? Could the author be more precise on this? Anyway, please write zinc compounds not components!!! The sentence “Consequently….as shown in Fig. 4(d) must be more clear and indicate also Fig. 4(e) and Fig. 4(f), otherwise the sentence is not complete.

6.     Results and discussion, figure 4: It is better to show the bars of measure inside the SEM images, instead of 80x, 300x and 5000x.

7.     Results and discussion, figure 5: This analysis is not clear. I cannot see the presence of Zinc in the porosity showed in Fig. 5(a). The author should present a more evident analysis. Performing a line scan analysis through the porosity could be more useful to evidence the presence of Zinc!!

8.     Results and discussion, figure 6: : It is better to show the bar of measure inside the SEM image.

9.     Results and discussion, table 5: In table 5 the author have reported the Porosity Ratio, so write in all the manuscript porosity ratio and not porosity rate. This makes confusion.

10. Results and discussion, figure 7: The conclusion of Fig. 7 explained at page 9 are quite superficial. It is obvious that if the higher the porosity rate, the lower the shear strength and if the shear strength is high it is because the porosity rate is low!! Moreover, in my opinion, the distribution of results in Fig. 7 don’t exactly show a clear tendency to have a reduction of TSS with increase of Porosity ratio. Maybe further tests should be made to clearly evidence this correlation. 

Author Response

Dear Reviewer

Sincerely appreciate your review of our paper in the midst of your busy schedule.

The comment of the reviewer is completed and attached as a word file.

Reviewer 2 Report

The manuscript has technical interest but not scientific concern.

The two conclusions are totally predictable:

-          welding a galvanized steel can lead to porosity defects induced by Zn

-          porosities decreased strength

In my opinion, the paper would be suitable for a journal devoted to technological aspects. In fact, most of the references given by the authors are taken from journals in the field of welding, such as Welding Int., J Welding Joining and Sci Technol Weld Join. I suggest to send the paper to one of these journals.

Author Response

(The authors gave the same response as above.)

Reviewer 3 Report

This research has studied the effect of gap between two coupons in lap-shear welded specimens. Through a systematic study it was shown that the specimens with no gap has significant level of porosity, while the specimens with 0.5mm gap have continuous weld with no porosity. The generation of porosities in specimens without gap is explained by the zinc fume, which is generated during the welding process, being trapped inside the melt pool. However, in samples with a small gap between the coupons the zinc fume is discharged, leaving the weld free of porosities. 

Below are my comments for the authors:

Review and cite this recently published paper in the introduction section: Kim, Dong-Yoon, et al. "Effect of Porosity on the Fatigue Behavior of Gas Metal Arc Welding Lap Fillet Joint in GA 590 MPa Steel Sheets." Metals 8.4 (2018): 241.

In section 2.1 introduce the testing equipment as well as experimental setup for the RT examination and the quasi-static tensile shear test.

It should be stated clearly how many specimens were cut from each of the welds shown in Table 3, and how far from each end was discarded to avoid the end effect. 

It was not explained why the welding parameters are different in the 6 runs listed in Table 3. 

In section 2.3, cite a reference for KS B 0801.

Adding a side view of the sample geometry is useful for Fig. 2.

The weld width "w" needs more explanation. Is it the projected width or the width of the weld on the fracture surface (which might be angled)? Alternatively, you can show "w" on a schematic of the welded specimen.

In section 3.1, after the first sentence, discuss if the same observation has been reported in the literature.

It is not scientific to show the level of magnification with a magnification factor. Use scale bar, instead in Fig. 5(a).

In section 3.2, explain what the "allowable tensile strength" is. Is it the yield strength of the base metal? What reference has introduced this as the allowable strength?

Explain how the TSS in Table 5 is calculated.

In Table 6, the fracture mode for run 2 is different than run 1 and 3 (fracture in HAZ vs. in the base metal). Discuss why?

Replace "porosity rate" with "porosity ratio" throughout the paper.

In section 3.2.1, in the sentence: "...results listed in Tables 3 and 4" it looks like that Tables 5 and 7 were meant to be referred to.

In section 3.2.1, authors should note that while the research shows that the gap (or lack of porosity) results in higher tensile shear strength, Fig. 7 shows that the correlation between porosity (%) and TSS is not really strong. 

The practical conclusion from this study is missing in the conclusion section: A gap of 0.5mm is an efficient way to discharge zinc fumes and prevents the formation of porosity in the weld.

Author Response

Dear Reviewer

Sincerely appreciate your review of our paper in the midst of your busy schedule.

The comment of the reviewer is completed and attached as a word file.

Round  2

Reviewer 1 Report

The modified version of the manuscript can be accepted for publication. I would just like to point out that equation (1) is not clearly written. Authors should modify it.

Author Response

Dear Reviewer

Sincerely appreciate your review of our paper in the midst of your busy schedule.

 - We modified the equation (1) according to the suggestion of the reviewer.

Weld Area (mm2) = Weld length (mm) x Weld width (mm)

Weld Area (mm2) = L (mm) x W (mm)

 Your comments and suggestions have been very helpful.

Thank you for your good comments.

Reviewer 2 Report

The paper can be accepted in the present form.

Author Response

Dear Reviewer

Sincerely appreciate your review of our paper in the midst of your busy schedule.

 Your comments and suggestions have been very helpful.

Thank you for your good comments.

Reviewer 3 Report

The paper is acceptable for publication, after editorial and English modifications.

Author Response

(The authors gave the same response as above.)
